# The Effect of SGLT2 Inhibition on Diabetic Kidney Disease in a Model of Diabetic Retinopathy

**DOI:** 10.3390/biomedicines10030522

**Published:** 2022-02-23

**Authors:** Jennifer Rose Matthews, Markus P. Schlaich, Elizabeth Piroska Rakoczy, Vance Bruce Matthews, Lakshini Yasaswi Herat

**Affiliations:** 1Dobney Hypertension Centre, School of Biomedical Sciences—Royal Perth Hospital Unit, University of Western Australia, Crawley, WA 6009, Australia; jen.matthews@uwa.edu.au (J.R.M.); vance.matthews@uwa.edu.au (V.B.M.); 2Dobney Hypertension Centre, Medical School—Royal Perth Hospital Unit, University of Western Australia, Crawley, WA 6009, Australia; markus.schlaich@uwa.edu.au; 3Department of Cardiology and Department of Nephrology, Royal Perth Hospital, Perth, WA 6000, Australia; 4Department of Molecular Ophthalmology, University of Western Australia, Crawley, WA 6009, Australia; elizabeth.rakoczy@uwa.edu.au

**Keywords:** diabetic kidney, sodium glucose co-transporter 2 inhibition, dapagliflozin, canagliflozin, empagliflozin, Akimba, Kimba, mouse model

## Abstract

Diabetic kidney disease (DKD) is a chronic disorder characterized by elevated urine albumin excretion, reduced glomerular filtration rate, or both. At present, angiotensin-converting enzyme inhibitors or angiotensin receptor blockers are the standard care for the treatment of DKD, resulting in improved outcomes. However, alternative treatments may be required because although the standard treatments have been found to slow the progression of DKD, they have not been found to halt the disease. In the past decade, sodium glucose co-transporter 2 (SGLT2) inhibitors have been widely researched in the area of cardiovascular disease and diabetes and have been shown to improve cardiovascular outcomes. SGLT2 inhibitors including canagliflozin and dapagliflozin have been shown to slow the progression of kidney disease. There is currently an omission of literature where three SGLT2 inhibitors have been simultaneously compared in a rodent diabetic model. After diabetic Akimba mice were treated with SGLT2 inhibitors for 8 weeks, there was not only a beneficial impact on the pancreas, signified by an increase in the islet mass and increased plasma insulin levels, but also on the kidneys, signified by a reduction in average kidney to body weight ratio and improvement in renal histology. These findings suggest that SGLT2 inhibition promotes improvement in both pancreatic and kidney health.

## 1. Introduction

Diabetic kidney disease (DKD) is a chronic disorder characterized by elevated urine albumin excretion, reduced glomerular filtration rate, or both. Globally, DKD is increasing in prevalence, with an estimated 500 million adults suffering from this disease, and is a major public health concern, especially in the low- to middle-income countries. DKD develops in approximately 40% of patients with type 2 diabetes (T2D) and 30% of patients with type 1 diabetes (T1D) and is the leading cause of chronic kidney disease (CKD) and end-stage renal disease [1,2]. The mortality risk associated with DKD has increased by 31.1% and increases with worsening disease severity [3]. It is reported that DKD affects males and females equally and it rarely develops before 10 years duration of T1D [4]. The humanistic, societal, and economic impact of DKD is enormous. It is an enormous burden on the health care system and seriously affects the physical health and quality of life of the patients [5]. In the United States in 2013 alone, kidney disease cost $50 billion dollars, with an additional $31 billion dollars spent on dialysis [6]. At present, angiotensin-converting enzyme (ACE) inhibitors or angiotensin receptor blockers (ARB’s) are the standard care for the treatment of DKD, resulting in improved outcomes. It has been shown that in an Italian diabetic population affected by both DKD and diabetic retinopathy, the role of multifactorial intervention achieved by using angiotensin converting enzyme inhibitors and angiotensin II receptor blockers in conjunction resulted in improved outcomes [7]. Although standard treatments have been found to slow the progression of CKD, they do not halt the disease. Therefore, alternative treatments may be required [8]. Other emerging therapies that are currently undergoing clinical trials include Endothelin Receptor A Antagonists, complement inhibition, Janus kinase (JAK) inhibition, chemokine inhibition [5], and the topic of our study, sodium glucose co-transporter 2 (SGLT2) inhibitors.

SGLT2 inhibitors are a novel class of antidiabetic medications currently approved (2013) by the Food and Drug Administration (FDA) for the treatment of diabetes [9]. SGLT2 is a high-capacity, low affinity glucose co-transporter, mainly found in the S1 and S2 segments of the renal convoluted proximal tubules, and is required for the reabsorption of majority of the glucose (~90–95%) filtered by the kidney [10,11]. In patients with diabetes mellitus, SGLT2 inhibitors increase glucosuria by blocking glucose reabsorption in the renal proximal tubule, and hence lower plasma glucose levels, independent of insulin stimulation [12].

In the past decade, SGLT2 inhibitors have been widely researched in the area of cardiovascular disease and diabetes and have been shown to improve cardiovascular outcomes. The mechanisms underpinning these effects remain incompletely understood but may include improved arterial stiffness and cardiac function, reduced cardiac oxygen demand as well as cardiorenal effects including a reduction in albuminuria and uric acid. 

SGLT2 inhibitors including canagliflozin (CANA) and dapagliflozin (DAPA) have been shown to slow the progression of kidney disease, as well as reduce the risk of kidney failure, as reported in the CREDENCE (Canagliflozin and Renal Events in Diabetes with Established Nephropathy Clinical Evaluation) [13] and DAPA-CKD [14] trials respectively. Currently, there is an omission of literature where three SGLT2 inhibitors have been simultaneously compared in a diabetic mouse model.

Here, we aimed to compare three SGLT2 inhibitors (dapagliflozin, empagliflozin (EMPA), and canagliflozin) and their effect on DKD in the well-characterized diabetic retinopathy mouse model, namely the Akimba mouse [15,16]. For the first time, the Kimba and Akimba mouse models were investigated for their kidney phenotype and were used to assess the impact of SGLT2 inhibition on DKD. 

## 2. Materials and Methods

### 2.1. Animals

Specific pathogen-free 10-week-old male Kimba and Akimba mice were obtained from the Animal Resources Centre (ARC, Perth, WA, Australia). All experimental and animal handling activities were performed at the Harry Perkins Institute for Medical Research animal holding facility (Perth, WA, Australia) according to the guidelines of the Institutional Animal Care and Use Committee. Animal ethics approval (AE141/2019, 12 February 2019) was received from the Harry Perkins Institute for Medical Research Animal Ethics Committee.

The Kimba mouse has transient photoreceptor-specific overexpression of human vascular endothelial growth factor 165 (VEGF165) and hence depicts changes associated with diabetic retinopathy without the hyperglycemic background [16]. The Akimba mouse (Ins2AkitaVEGF+/−) is a cross between the diabetic Akita [17] and the Kimba model [16], and develops advanced diabetic retinopathy features due to the interplay between high blood glucose levels and overexpression of VEGF [15].

All mice were acclimatized for a period of seven days. Mice were housed under a 12-h light/dark cycle at 21 ± 2 °C and were given a standard chow diet (Specialty Feeds, Glen Forrest, WA, Australia) with free access to food and drinking water containing the SGLT2 inhibitor (DAPA, EMPA, or CANA; Ark Pharma Scientific Limited, Wuhan, China; 25 mg/kg/day) or Vehicle (dimethylsulfoxide). Wet mash was replenished three times per week. Mice underwent treatment for a period of 8 weeks. Drinking water containing the SGLT2 inhibitor or Vehicle was freshly prepared and replaced on a weekly basis. To establish the diabetic status in Akimba mice, urine glucose levels were measured (Keto-Diastix; Bayer, Leverkusen, Germany) before treatment. In non-diabetic Kimba mice, the development of glucosuria induced by the particular SGLT2 inhibitor was measured 1-week post-treatment. Mice were weighed weekly.

### 2.2. Tissue Sample Collection and Processing

At the end of the experiment, mice were anesthetised using isoflurane inhalation. Mice were sacrificed and the kidneys were collected and weighed. The left kidney was fixed for 24 h in 10% buffered formalin for histology and the right kidney was snap frozen and stored at −80 °C for subsequent experiments. Pancreatic tissue was collected and fixed for 24 h in 10% buffered formalin for histology. Blood was collected by cardiac puncture and serum was collected.

### 2.3. Assessment of Renal Parameters

To determine renal hypertrophy, both kidneys were weighed and the average kidney to body weight ratio (K/BW Ratio) was calculated.

### 2.4. Histology of Kidney and Pancreatic Tissues

After fixation, pancreatic and kidney tissues were paraffin-embedded and sectioned at 5µm thickness using a Leica semi-automated RM2245 microtome (Leica Biosystems, Sydney, NSW, Australia). Sections were used for hematoxylin and eosin, Periodic acid-Schiff (PAS), and Masson’s trichrome staining for morphometric analysis. In PAS staining, saccharides will be stained red and the nucleus will be stained blue. Collagen will be stained blue, the nucleus will be stained dark purple, while fiber will be stained red in Masson’s trichrome.

### 2.5. Immunohistochemistry of Kidney Tissues

The detection of SGLT2 protein expression in kidney tissue from Kimba and Akimba mice was conducted as published previously [18]. Briefly, kidney tissue was fixed in 10% buffered formalin for 24 h, followed by wax embedding. Paraffin sections (5 μm) were collected and mounted on slides. For antigen retrieval, slides were heated for 2.5 min in a pre-heated 1× EDTA buffer (pH 8.5; Sigma-Aldrich, Sydney, NSW, Australia). After washing twice in PBS/0.1% Tween for 5 min, tissue sections were outlined with a paraffin pen. Sections were blocked with 3% H_2_O_2_ for 10 min, washed twice with PBS/0.1% Tween for 5 min, and blocked with 5% FBS in PBS/0.1% Tween for 1 h in a humidified chamber. Sections were then incubated overnight at 4 °C in a humidified chamber with both rabbit anti-SGLT2 (1:200; Novus Biologicals, Littleton, CO, USA) and mouse anti-SGLT2 antibodies (1:50; Santa Cruz Biotechnology, Santa Cruz, CA, USA) in 5%FCS/PBS/0.1% Tween. Following overnight incubation, sections were washed three times with PBS/0.1% Tween for 5 min and incubated with anti-rabbit (1:100, Santa Cruz Biotechnology, Sydney, NSW, Australia) and anti-mouse (1:100, Santa Cruz Biotechnology, Sydney, NSW, Australia) secondary antibodies conjugated with HRP in PBS/0.1% Tween for 1 h. This was followed by incubation with diaminobenzidine (DAB). Slides were counterstained with hematoxylin, dehydrated, and mounted with DPX (Sigma-Aldrich, Sydney, NSW, Australia).

### 2.6. Western Blotting

Murine kidney tissue was homogenized using cytosolic extraction buffer containing phosphatase and protease inhibitors, and protein concentration was quantified using protein assay solution (Bio-Rad, Hercules, CA, USA). Protein lysates were resolved by SDS-PAGE on 10% polyacrylamide gels, transferred by semi-dry transfer to PVDF membrane. Membranes were incubated overnight in primary antibody (anti-SGLT2 (sc-393350; Santa Cruz Biotechnology, Dallas, TX, USA) or anti-β actin antibody (ab6276; Abcam, Cambridge, UK) using recommended dilutions. The appropriate secondary antibody was added to the membranes [α-mouse 680 (LICOR)] for Odyssey detection. Detection of the relevant protein was visualized using an Odyssey detection apparatus (LICOR).

### 2.7. Imaging and Quantitation

The H&E-stained pancreatic and kidney tissues and SGLT2 stained kidney tissues were visualized and imaged using the inverted Nikon Eclipse Ti microscopic system (Nikon, Tokyo, Japan) and a CoolSNAP HQ2 digital camera (Photometrics, Tucson, AZ, USA). 

The mean SGLT2 staining intensity score of stained tubules was calculated in counterstained kidney tissue in 5 random fields of view. The SGLT2 staining intensity was scored on a scale of 0–3 (0 = no staining; 1 = low; 2 = intermediate; 3 = high).

### 2.8. Statistical Analysis

Data was analyzed using a two-tailed Student’s *t*-test or one-way ANOVA as appropriate. Quantitative data is presented as mean ± SEM. Data was deemed significant when *p* < 0.05. Graphs were produced with GraphPad Prism 8 (GraphPad Software Inc., San Diego, CA, USA).

## 3. Results

### 3.1. SGLT2 Inhibition Promotes Glucosuria in Non-diabetic Kimba Mice

The non-diabetic Kimba mouse model does not normally excrete glucose in the urine. Therefore, in order to ascertain the bioactivity of the SGLT2 inhibitors, after 3 days of treatment, the urine of Kimba mice were tested (Figure 1) for glucosuria. As expected, treatment with all three inhibitors promoted glucosuria, with DAPA producing the most profound effect.

### 3.2. SGLT2 Inhibition Promotes Weight Gain and the Ability to Thrive in Akimba Mice

At the end of the 8-week treatment regime, non-diabetic Kimba mice were significantly higher than diabetic Akimba mice. In addition, all SGLT2 inhibitors promoted weight gain in diabetic animals when compared to vehicles. In particular, a significant increase in weight gain was noted with the SGLT2 inhibitors EMPA and CANA when compared to vehicle treated animals (Figure 2). 

### 3.3. SGLT2 Inhibition may Improve Pancreatic Health in Diabetic Akimba Mice

Pancreata were examined using H&E staining. Diabetic Akimba mice demonstrated increased islet mass after SGLT2 inhibitor treatment with EMPA and CANA (Figure 2), suggesting an improvement in pancreatic health. A histologic assessment of general islet structure indicated that pancreatic islet mass was affected by the progression of hyperglycemia in diabetic Akimba mice, as demonstrated by the appearance of small and irregular islet architecture (Figure 3B,F,J), whereas normal islet architecture and mass was preserved in the non-diabetic Kimba mice (Figure 3A,E,I). Diabetic Akimba mice demonstrated increased islet mass following 8 weeks of SGLT2 inhibitor treatment with EMPA (Figure 3C,G,K) or CANA (Figure 3D,H,L), as compared with Akimba Vehicle controls. As the SGLT2 inhibitor EMPA increased islet mass, we aimed to ascertain the effect of EMPA on serum insulin levels in diabetic Akimba mice. Non-diabetic Kimba mice treated with EMPA showed significantly increased plasma insulin levels compared to Vehicle treated counterparts. Although not significant, an elevation in plasma insulin levels were noted in Akimba mice treated with EMPA compared with the Vehicle group (Figure 4).

### 3.4. SGLT2 Inhibition Improved Renal Hypertrophy in Diabetic Akimba Mice

In an effort to garner whether SGLT2 inhibition promotes benefits on kidney health in diabetic Akimba mice, we assessed the average kidney to body weight (K/BW) ratio after 8 weeks on SGLT2 inhibitor therapy. When compared to Vehicle-treated non-diabetic Kimba mice, the diabetic Akimba mice treated with Vehicle showed a significant increase in the K/BW ratio indicating renal hypertrophy in Akimba mice. Diabetic Akimba animals treated with either DAPA or CANA displayed significant reductions in the K/BW ratio compared to Vehicle treated diabetic Akimba mice. However, Akimba mice treated with EMPA showed no change in the K/BW ratio when compared to the Vehicle treated counterparts (Figure 5). 

### 3.5. SGLT2 Inhibition with DAPA Promotes Compensation of SGLT2 in the Kidneys of Diabetic Akimba Mice

After SGLT2 immunohistochemistry was conducted on the kidneys of the diabetic Akimba mice, the intensity of SGLT2 tubule staining was assessed. It was found that compared to Vehicle, DAPA resulted in a significant increase in SGLT2 protein expression in the kidneys of Akimba mice (Figure 6), indicating a compensation of SGLT2 protein expression (Figure 6). In agreement with the SGLT2 immunohistochemistry, the specific 77 kDa SGLT2 protein is elevated in kidney of Akimba mice treated with DAPA (Figure 6D). However, EMPA (Figure 7) and CANA (Figure 8) did not demonstrate an increase of SGLT2 protein expression.

### 3.6. The Effect of SGLT2 Inhibition on Renal Histology in Diabetic Akimba Mice

In DKD, common renal histopathologic findings include prominent mesangial expansion, tubular dilation, and tubulointerstitial fibrosis. Renal morphometric analysis demonstrated that renal tubules and glomeruli appeared normal in non-diabetic Kimba mice treated with Vehicle (Figure 9A–C). However, Vehicle-treated Akimba mice demonstrated presence of tubular dilation, degeneration and vacuolation of renal tubules, tubulointerstitial fibrosis, mesangial expansion, and shrunken glomeruli with wide bowman spaces, indicative of renal damage caused by diabetes (Figure 9D–F). Akimba mice treated with EMPA demonstrated marked improvement in renal histology (Figure 9G–I), indicating the potential beneficial effect of EMPA in improving renal health. Furthermore, Masson’s trichrome staining (Figure 10) revealed significant renal interstitial fibrosis (dyed blue) in diabetic Akimba group (Figure 10D–F) compared to Akimba mice treated with EMPA (Figure 10G–I). The major pathological alterations observed in PAS-stained (Figure 11) diabetic Akimba kidneys include glomerular hypertrophy, basement membrane thickening, and mesangial matrix deposition (Figure 11D–F), which significantly improved with EMPA treatment. 

## 4. Discussion

Diabetic Kidney Disease (DKD) is increasing in prevalence amongst both T1D and T2D individuals and hence, therapies are urgently needed. Due to the side effects of standard therapies such as ACE inhibitors and ARB’s, SGLT2 inhibitors for the treatment of DKD has become a promising therapeutic approach. 

The SGLT2 inhibitor drug class has been shown to exert anti-inflammatory actions by suppressing the activation of P3 receptor inflammasome activity (NLRP3), anti-fibrotic affects by stimulating M2 macrophages and inhibiting myofibroblast differentiation [19] and improves endothelial dysfunction [20]. In addition, an important method SGLT2 inhibitors promote nephroprotection is by tubuloglomerular feedback, in which SGLT2 inhibitors cause more sodium to pass along the nephron. This influx of sodium is sensed by macula cells to constrict afferent glomerular arterioles, and this reduces blood flow, thereby protecting glomeruli by reducing intraglomerular pressure [11]. These mechanisms support the synergistic effect exerted by these medications and together could explain the benefits of SGLT2 inhibitors in DKD.

In this study, we investigated the impact of SGLT2 inhibition on kidney health in the novel strain of diabetic Akimba mice. To our knowledge, this is the first study to investigate the diabetic kidney disease phenotype in the Akimba mice and to determine the effect of SGLT2 inhibition on the kidney health of this TID mouse model.

While all three SGLT2 inhibitors were effective in promoting urinary glucose excretion, DAPA in particular showed a more profound effect on urine glucose excretion (Figure 1), and this resulted in a concomitant lowering of blood glucose levels (unpublished data). Similarly, a study by Tahara et al. in a T2D murine model has shown a comparable increase in urinary glucose excretion and a reduction in blood glucose levels with DAPA when compared to the other SGLT2 inhibitors EMPA and CANA, used in this study [21]. The persistence of pharmacologic effects of SGLT2 inhibitors has been confirmed to be strongly dependent on their pharmacokinetics, particularly on their distribution and retention in the target organ, the kidney [22]. While DAPA is considered a long-acting SGLT2 inhibitor, EMPA and CANA are considered intermediate-acting based on the pharmacokinetic, pharmacodynamic, and pharmacologic results [22]. The results of the present study support the superiority of the long-acting SGLT2 inhibitors in relation to their anti-diabetic effects.

A wealth of studies in both T1D and T2D animal models have demonstrated that SGLT2 inhibition improves beta cell function and maintains both beta cell mass and islet morphology [23,24,25,26,27,28]. The present study has demonstrated, for the first time, the protective role of EMPA in the improvement of islet mass and insulin secretion in the diabetic Akimba mouse model. In Akimba mice treated with Vehicle, the islets were atrophied; however, when those mice were given either the SGLT2 inhibitors EMPA or CANA, there was an improvement in the size and health of the pancreatic islets which was comparable to the non-diabetic pancreas of Kimba mice (Figure 3). This observation is further supported by the results of previous studies in other T1D rodents in which EMPA alone improved pancreatic parameters and glucose homeostasis [23,29]. 

Although various SGLT2 inhibitors have been shown to have significant protective effects on the kidneys and cardiovascular system in patients with diabetes, recent data has shown that DAPA is also beneficial for those individuals with non-diabetic kidney disease [30]. Therefore, it has recently been approved as a therapy to reduce the risk of declining kidney function and kidney failure, with and without diabetes [31]. Renal hypertrophy is a characteristic in early DKD arising from uncontrolled diabetes. Furthermore, tubule dilation, fibrosis, and enlarged glomeruli are markers of diabetes-associated kidney damage [32]. SGLT2 inhibitors exert renoprotective effects by suppressing several processes associated with kidney diseases, such as increased kidney weight, fibrosis, and mesangial expansion [33,34,35,36]. Renal hypertrophy was observed in Vehicle-treated diabetic Akimba animals, which was evident by increased K/BW ratio (Figure 5). We showed that treatment with DAPA and CANA significantly improved the K/BW ratio owing to an improvement in renal hypertrophy, although EMPA did not demonstrate a similar improvement. However, in this study, EMPA improved the real disease pathology, which was evident by renal histology (Figure 9, Figure 10 and Figure 11), and Domon et. al. has reported similar findings in a rat model of diabetes with enlarged kidney disease [37]. Interestingly, we recently demonstrated an improvement in renal hypertrophy with young Akimba mice (~5-week-old) treated with EMPA [38]. This demonstrates that, although an improvement in renal hypertrophy was not evident in older Akimba mice with the treatment of EMPA, it does demonstrate the ability to promote benefits in renal phenotype and this warrants further investigations which include functional assays.

In this study, we looked at whether the three commonly used SGLT2 inhibitors DAPA, EMPA, and CANA increased SGLT2 protein expression in the kidneys of diabetic Akimba mice. We have previously shown that DAPA treatment significantly increased the number of proximal tubules expressing SGLT2 in C57BL6/J mice fed a high-fat diet [13]. Similarly, in this study, DAPA was the only SGLT2 inhibitor that demonstrated an increase in expression of SGLT2, suggestive of a compensatory increase in the presence of SGLT2 inhibition. Although upregulation of SGLT2 occurs when DAPA is administered, the SGLT2 inhibitors will continue to reduce the activity of the newly formed SGLT2 protein. Therefore, whether the compensatory increase in SGLT2 protein is biologically relevant requires further exploration. 

Recent clinical studies highlight the important benefits of SGLT2 inhibition in T1D. The EASE (Empagliflozin as Adjunctive to inSulin thErapy)-1 and -2 programs investigated EMPA in T1D subjects and showed a clear beneficial effect on HbA1c, body weight, blood pressure, glucose variability, and total daily insulin use with minimal adverse events [39,40]. The EASE-3 program only used a very low dose of Empagliflozin (2.5 mg) but still revealed beneficial effects on metabolic and blood pressure outcomes includingHbA1c, body weight, total daily insulin dose, and systolic blood pressure. Interestingly, a lower risk of hypoglycemia and diabetic ketoacidosis was observed. These metabolic benefits suggest a novel role for EMPA in the setting of T1D [40]. At the kidney level, alterations in sodium handling may also drive the observed renal hemodynamic actions, where more distal sodium uptake could drive diabetic hyperfiltration suppression independent of effects on blood glucose lowering [41].

In addition, we have evidence that reducing SGLT2 activity results in upregulation of renal SGLT1 (manuscript in preparation). SGLT1 is another sodium-dependent glucose cotransporter found in the S3 (distal) segment of the proximal tubules. This highlights the necessity to inhibit both SGLT1 and SGLT2 with the currently available dual SGLT1/2 inhibitor Sotagliflozin [42].

Our current study highlights that SGLT2 inhibitors not only improve pancreatic health but also improve kidney health in our diabetic Akimba mouse. Of major significance, our study is the first kidney related study performed in Akimba mice and it is also the first study to show the beneficial effects of CANA on the pancreas.

Despite being able to investigate the Kimba and Akimba mouse models for their kidney phenotype, our study does have certain limitations. Firstly, the tissues used in this study was initially designed to determine the effect of SGLT2 inhibition on diabetic retinopathy. The diabetic retinopathy studies were conducted on male animals as the disease progression in females is slower and less uniform [15,16]. Given diabetic retinopathy is associated with DKD [43] we investigated the kidney phenotype in our male mice. Although the retinal phenotype may vary in female mice, the kidney phenotype should be assessed in females in future studies to determine if the renal phenotypes different by sex. Secondly, we investigated only the concertation of 25 mg/kg/day given for a period of 8 weeks via drinking water for all treatments. It has been shown that SGLT2 inhibitors have a dose-dependent effect on several metabolic parameters and reno-protective effects in rodents [35,36,44]. Therefore, future studies should investigate the effect of a higher dose of the SGLT2 inhibitors and its effect on DKD in our mouse models. In addition, alternative time course regimes should also be evaluated. Lastly, the present study was conducted on 10-week-old Kimba and Akimba mice. The retinal changes in Kimba and Akimba mice represents moderate changes in severity at this age and hence the kidney phenotype observed could also be moderate in nature. Our current investigations focus on the effect of SGLT2 inhibition on the progression of DKD. Therefore, it is likely that younger animals could demonstrate a less severe disease renal phenotype and could be used to determine if SGLT2 inhibitors are able to halt the development of DKD. These experiments will be performed in future studies.

In addition, future studies in Kimba and Akimba mice should: (1) encompass the measurement of plasma and tissue concentrations of SGLT2 inhibitors using high-performance liquid chromatography, (2) measure blood pressure of the animals to determine the effect of SGLT2 inhibition on blood pressure and renal phenotype in these models, (3) accurately measure the effects of SGLT2 inhibition on urine volume and urinary sodium excretion using metabolic cages, and (4) the urinary albumin creatinine ratio (uACR) and blood urea nitrogen levels should be measured to further characterize the Kimba and Akimba mouse models and to determine the beneficial effect of SGLT2 inhibition on these parameters. 

## 5. Conclusions

In this study we showed that SGLT2 inhibitors not only promoted glucosuria in non-diabetic Kimba mice but also improved pancreatic health and renal hypertrophy in diabetic Akimba mice, as well as DAPA promoting compensation of SGLT2 in the kidneys of these diabetic mice.

## Figures and Tables

**Figure 1 biomedicines-10-00522-f001:**
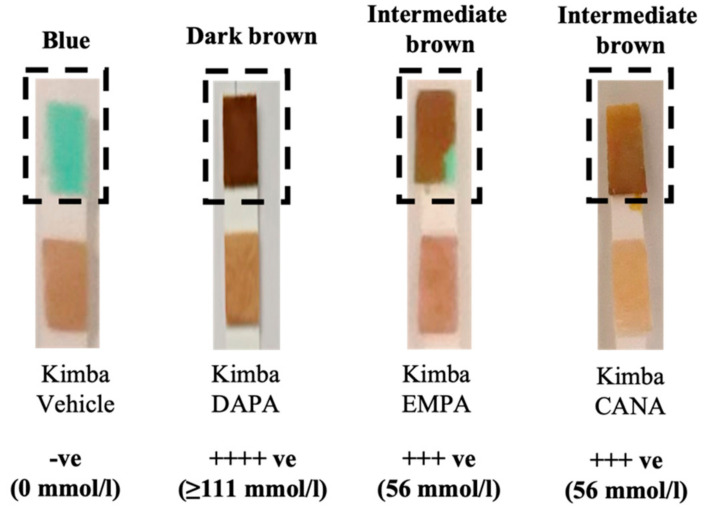
SGLT2 inhibition with DAPA, EMPA, and CANA promotes glucosuria in non-diabetic Kimba mice. DAPA: Dapagliflozin; EMPA: Empagliflozin, and CANA: Canagliflozin.

**Figure 2 biomedicines-10-00522-f002:**
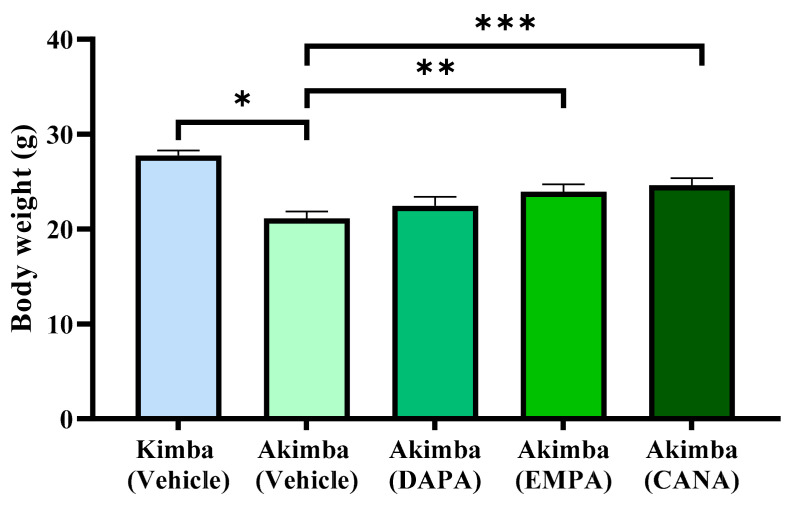
SGLT2 inhibition prevents failure to thrive phenotype in diabetic Akimba mice. Final body weight values are expressed as means + SEM; *n* = 3−18 mice/group. * *p* < 0.0001 (Kimba Vehicle vs. Akimba Vehicle); using One-way ANOVA ** *p* < 0.05 (Akimba EMPA vs. Vehicle); *** *p* < 0.01 (Akimba CANA vs. Vehicle). DAPA: Dapagliflozin; EMPA: Empagliflozin and CANA: Canagliflozin.

**Figure 3 biomedicines-10-00522-f003:**
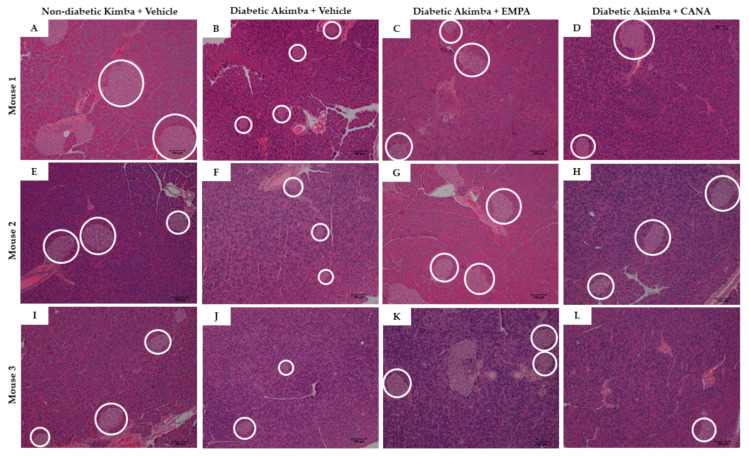
H&E staining of pancreatic tissue from Kimba mice treated with Vehicle (**A**,**E**,**I**) and Akimba mice treated with Vehicle (**B**,**F**,**J**), EMPA (**C**,**G**,**K**), and CANA (**D**,**H**,**L**). White circles indicate islets. Scale bar = 100 µm; Magnification = 100×. Three representative mice per treatment. H&E: Hematoxylin & eosin; EMPA: Empagliflozin; and CANA: Canagliflozin.

**Figure 4 biomedicines-10-00522-f004:**
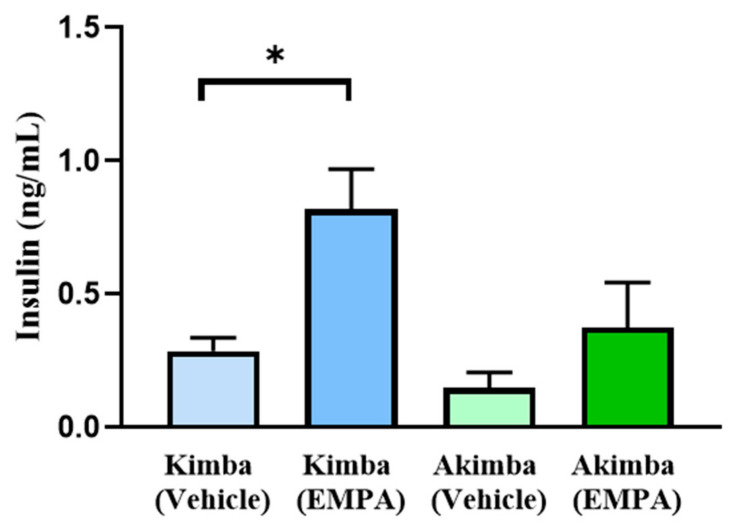
SGLT2 inhibition with EMPA increases circulating insulin levels in both Kimba and Akimba mice. Data represented as mean + SEM, *n* = 5–6 mice/group (Kimba), *n* = 3 mice/group (Akimba), using One-way ANOVA * *p* < 0.01 (Kimba Vehicle vs. Kimba EMPA). EMPA: Empagliflozin.

**Figure 5 biomedicines-10-00522-f005:**
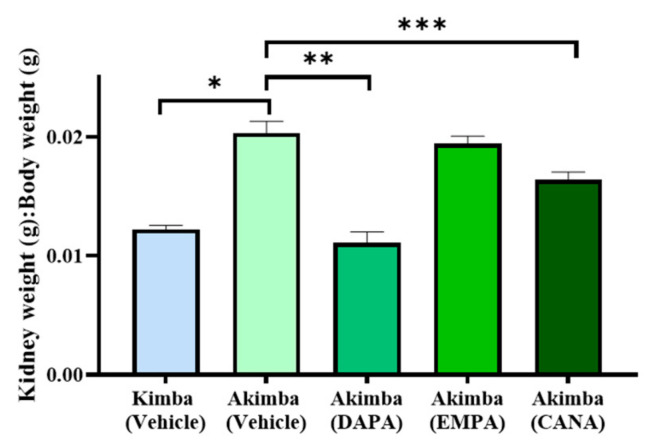
Effect of SGLT2 inhibition on kidney to body weight ratio (KW/BW) in diabetic Akimba mice. Data represented as mean + SEM, *n* = 3–18 mice/group. Using One-way ANOVA * *p* < 0.0001 (Kimba Vehicle vs. Akimba Vehicle); ** *p* ≤ 0.0001 (Akimba DAPA vs. Vehicle); *** *p* ≤ 0.006 (Akimba CANA vs. Vehicle). DAPA: Dapagliflozin; EMPA: Empagliflozin and CANA: Canagliflozin.

**Figure 6 biomedicines-10-00522-f006:**
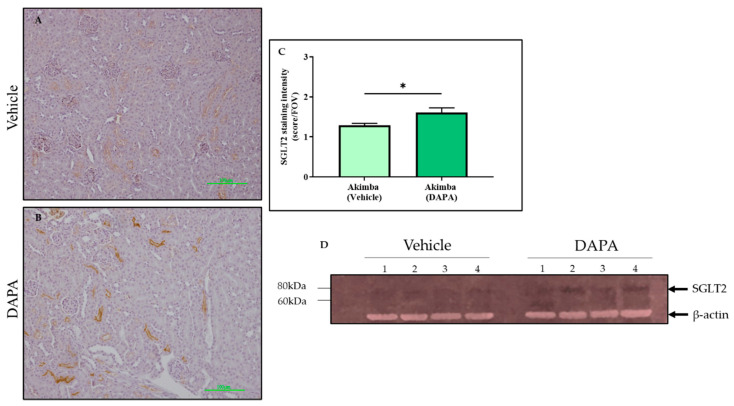
DAPA increases SGLT2 protein expression in the kidneys of diabetic Akimba mice. Representative images of SGLT2-stained kidney tissues from diabetic Akimba mice treated with Vehicle (**A**) and diabetic Akimba mice treated with DAPA (**B**), SGLT2 quantification (**C**) and immunoblots for SGLT2 and b-actin (**D**). Scale bar = 100 µm; *n* = 4 mice/group; using student’s *t*-test * *p* ≤ 0.05; DAPA: Dapagliflozin.

**Figure 7 biomedicines-10-00522-f007:**
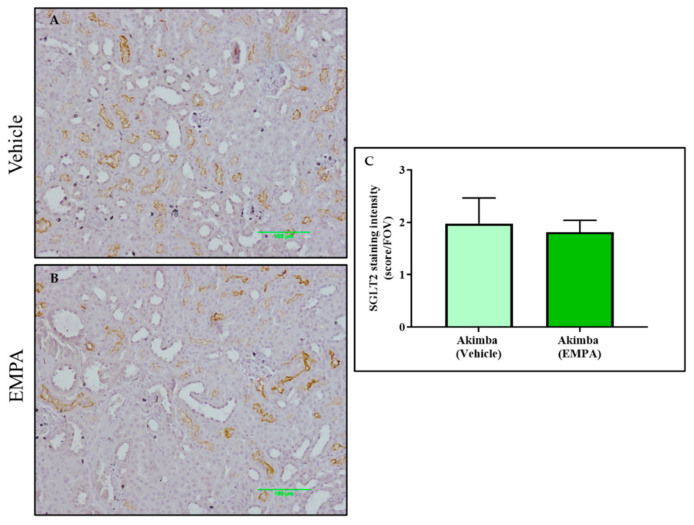
EMPA was not associated with increased SGLT2 protein expression in the kidneys of diabetic Akimba mice. Representative images of SGLT2-stained kidney tissues from diabetic Akimba mice treated with Vehicle (**A**) and diabetic Akimba mice treated with EMPA (**B**) and SGLT2 quantification (**C**). Scale bar = 100 µm; *n* = 3 mice/group; EMPA: Empagliflozin.

**Figure 8 biomedicines-10-00522-f008:**
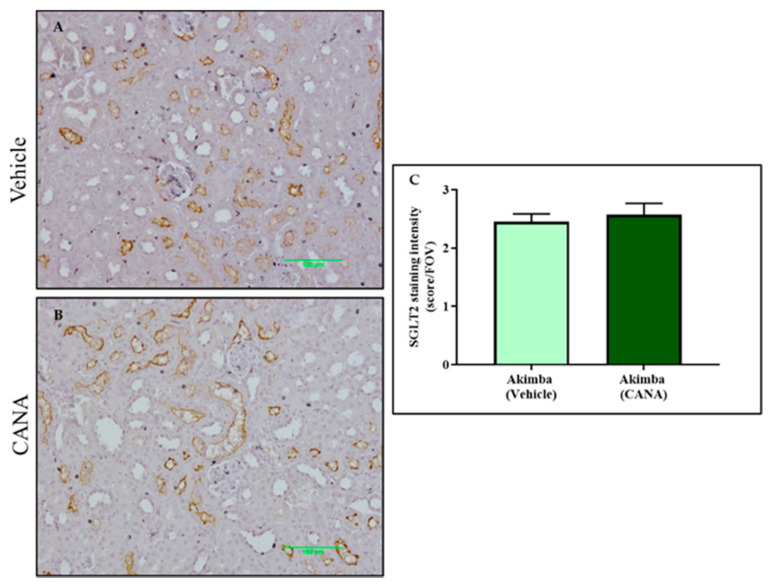
CANA was not associated with an increased SGLT2 protein expression in the kidneys of diabetic Akimba mice. Representative images of SGLT2-stained kidney tissues from diabetic Akimba mice treated with Vehicle (**A**) and diabetic Akimba mice treated with CANA (**B**) and SGLT2 quantification (**C**). Scale bar = 100 µm; *n* = 5–6 mice/group; CANA: Canagliflozin.

**Figure 9 biomedicines-10-00522-f009:**
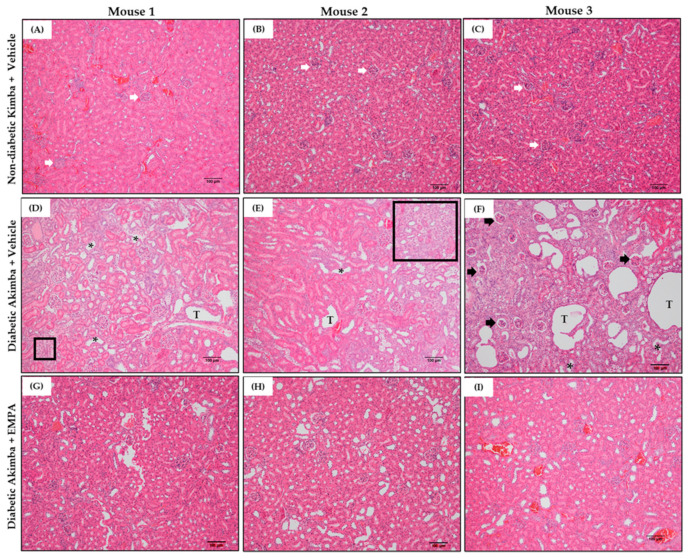
Representative images of H&E-stained kidney tissues from non-diabetic Kimba mice treated with Vehicle (**A**–**C**), diabetic Akimba mice treated with Vehicle (**D**–**F**) and diabetic Akimba mice treated with EMPA (**G**–**I**). Glomeruli (**A**–**C**; white arrows), tubular dilation (**D**–**F**; black asterisks), degeneration, and vacuolation of renal tubules (**D**–**F**; letter **T**), tubulointerstitial fibrosis (**E**; black box) and shrunken glomeruli with wide Bowman’s spaces (**F**; black arrows) are shown. Scale bar = 100 µm; Magnification = 100×; representative images are shown for 3 mice/group. EMPA: Empagliflozin.

**Figure 10 biomedicines-10-00522-f010:**
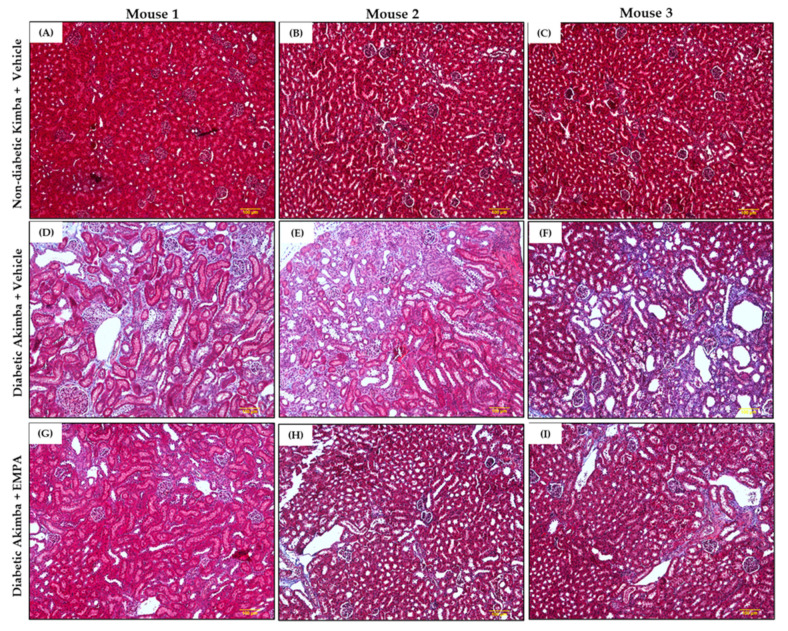
Representative images of Masson’s Trichrome-stained kidney tissues from non-diabetic Kimba mice treated with Vehicle (**A**–**C**), diabetic Akimba mice treated with Vehicle (**D**–**F**) and diabetic Akimba mice treated with EMPA (**G**–**I**). Collagen fibers stained blue in diabetic kidneys indicating renal interstitial fibrosis. Scale bar = 100 µm; Magnification = 100×; representative images are shown for 3 mice/group. EMPA: Empagliflozin.

**Figure 11 biomedicines-10-00522-f011:**
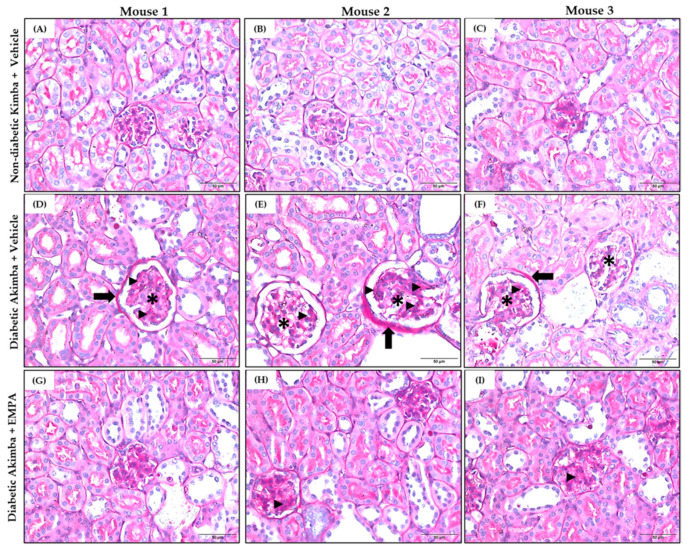
Representative images of Masson’s Trichrome-stained kidney tissues from non-diabetic Kimba mice treated with Vehicle (**A**–**C**), diabetic Akimba mice treated with Vehicle (**D**–**F**) and diabetic Akimba mice treated with EMPA (**G**–**I**). Glomerular hypertrophy (**D**–**F**; black asterisks), basement membrane thickening (**D**–**F**; black arrows), and mesangial matrix deposition (**D**–**F**,**H**,**I**; black arrow heads). Scale bar = 50 µm; Magnification = 400×; representative images are shown for 3 mice/group. EMPA: Empagliflozin.

## Data Availability

Not applicable.

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
