# Peer review of "The Effect of SGLT2 Inhibition on Diabetic Kidney Disease in a Model of Diabetic Retinopathy"

_biomedicines, 2022, doi:10.3390/biomedicines10030522_

Round 1

Reviewer 1 Report

Thanks for answering the remaining concerns bout your paper.

Author Response

Please see the attachment with the point-by-point response to Reviewer 1.

Reviewer 2 Report

The article is very interesting, once Sodium Glucose Transporter 2 inhibitors (SGLT2i’s) have been widely researched in the area of cardiovascular disease and diabetes and have been shown to improve cardiovascular outcomes. The article is very well written and clear.

General comments:                        

  1. The abstracted is good.
  2. The introduction and the discussion could be improved, with a better explanation of SGLTe (https://link.springer.com/article/10.1007/s11892-021-01442-z). This SGLT2-i are the new class of antidiabetic medications which are recently approved (2013) by the Food and Drug Administration (FDA) for the treatment of diabetes (DOI: 10.2174/1574886313666180226103408).
  3. The data are well explained and the statistics are well applied.
  4. The conclusion are good.

Author Response

Please see the attachment with the point-by-point response to Reviewer 2.

This manuscript is a resubmission of an earlier submission. The following is a list of the peer review reports and author responses from that submission.

Round 1

Reviewer 1 Report

The paper “The Effect of SGLT2 Inhibition on Diabetic Kidney Disease in 2 a Model of Diabetic Retinopathy" by Matthews et al. is a study with the aim of comparing three SGLT2i’s (dapagliflozin, empagliflozin and canagliflozin) and to assess their effect on DKD in the well-characterized diabetic retinopathy mouse model.

 The article is well written, and only minor spell check is needed. The study has a good design. The article is logically divided into sections and subsections. There are several tables and figures of good quality. The references cited are relevant and adequate. The work has an average degree of novelty and of good interest to the readers.

Comments:

  • Line 52: an Italian randomized clinical trial, with a long follow up, showed that in a diabetic population affected by both DKD and retinopathy, the role of multifactorial intervention, also achieved by using together ACEi’s and ARB’s, resulted in improved outcome.  
  • Discussion: SGLT2i beneficial effects on kidney should be improved by adding the beneficial anti-inflammatory and anti-fibrotic effect of this class drug (doi: 10.3390/ijms22115863), as well as their role on endothelial disfunction (doi: 10.3390/biomedicines9101356).

Reviewer 2 Report

SGLT2i’s are of great interest and as mentioned in the paper not only in the area of cardiovascular disease and diabetes but also in chronic kidney disease.  Therefore it is very important and of great interest to investigate SGLT2 inhibitors in chronic kidney disease.  This study aimed to evaluate and compare the three SGLT2i’s (dapagliflozin, empagliflozin and canagliflozin) in the Akimba diabetic retinopathy mouse model.

Figure 3:

Are there any explanation why the insulin levels are increased in the EMPA treated mice?

Figure 4:

The avarage weight between the different groups would be of interest; maybe the mice just were more starving and lost weight or gained even weight?

Figure 5- 7:

Unfortunately, the Immunohistochemistry staining’s for DAPA seems to be quite unspecific.

To underline the statement that SGLT2 protein expression in the kidneys of Akimba mice in EMPA, CANA and DAPA mice is different a Wetsernblot should be performed.

Since SGLT2 is localized at the proximal tubule, Matthews et al should also perform a co-staining with sglt2 and  a proximal tubule marker.  Just a side note, the scale bars of Figure 5/6/7 are different. In addition, the tubules seems to be more dilated in the EMPA and CANA kidneys compared to the DAPA kidney tissue.  Is there any explanation for that?  Also in Figure 8 the scale bars are of different colour and different intensity of the staining.

Figure 8:

It is postulated that Akimba mice treated with EMPA have an improved phenotype in renal histology (Figure 8 G-I), but no further staining’s like PAS to investigate sclerosis or fibrotic staining (periodic acid–Schiff or Sirius Red) was performed.

Other open questions:

As the authors claim to investigate a diabetic kidney, disease mouse model the kidney phenotype should be evaluated in more detail. Are those mice proteinuria/Albuminuria? Are those mice have an increase in BUN?

Conclusions with a number of 3 mice per group are too less and are therefore cannot be accurately.

Since it is known that SLGT2s also have glucose-independent effects, such as blood pressure–lowering ones it would be of interest if the mice treated with the different SGLT2s have differences in their blood pressure values.

Since the drinking water was containing the SGLT2 inhibitors and mice could drink ad libitum, plasma levels in the mice of DAPA, EMPA, CANA would be of interest to see if all of the mice had the same values. Moreover, it could as well also not be excluded that the different groups have different urine output. Therefore, urine output in 24 hours should be measured.

Reviewer 3 Report

The subject of the article is very important, once DKD is increasing in prevalence amongst both T1DM and T2DM individuals and therapies are urgently needed. The article showed that SGLT2 inhibitors not only promoted glucosuria in non- diabetic Kimba mice but also improved pancreatic health and renal hypertrophy in diabetic Akimba mice, as well as DAPA promoting compensation of SGLT2 in the kidneys of these diabetic mice.

 General comments:

  1. The abstracted is good and consistent.
  2. The introduction is focused and addresses the main important items about the pathology and the study. However, there are some undefined abbreviations (e.g. CREDENCE, EMPA)
  3. In the objective of the work the author place a citation “Here, we aimed to compare three SGLT2i’s (dapagliflozin, empagliflozin and canagliflozin) and their effect on DKD in the well-characterized diabetic retinopathy mouse model, namely the Akimba mouse [10,11].” Why??
  4. There is no scale on some pictures, please enter.
  5. In the figures enter the statistical test used in the legend of the figure.
  6. The discussion could be improved by further exploring the molecular and mechanistic components of diabetes and drugs. There is already work the mechanism underlying the potential benefits of SGLT2 inhibition in T1D and its complications. so I think the discussion can be greatly improved.
  7. The limitations of the study should be included.